# Molecular and Antimicrobial Susceptibility Characterization of *Escherichia coli* Isolates from Bovine Slaughterhouse Process

**DOI:** 10.3390/antibiotics12020291

**Published:** 2023-02-01

**Authors:** José Vázquez-Villanueva, Karina Vázquez, Ana Verónica Martínez-Vázquez, Alfredo Wong-González, Jesus Hernández-Escareño, Omar Cabrero-Martínez, Wendy Lizeth Cruz-Pulido, Abraham Guerrero, Gildardo Rivera, Virgilio Bocanegra-García

**Affiliations:** 1Centro de Biotecnología Genómica of Instituto Politécnico Nacional, Reynosa 88710, Mexico; 2Facultad de Medicina Veterinaria y Zootecnia, Universidad Autónoma de Nuevo León, Escobedo 66050, Mexico; 3Campus Reynosa, Universidad del Valle de México, Reynosa 88760, Mexico; 4CONACyT Research, Centro de Investigación en Alimentación y Desarrollo, Mazatlán 82112, Mexico

**Keywords:** *Escherichia coli*, slaughterhouse, resistance, virulence, phylogroups

## Abstract

Antimicrobials are routinely used in human and veterinary medicine. With repeated exposure, antimicrobials promote antibiotic resistance, which poses a threat to public health. In this study, we aimed to determine the susceptibility patterns, virulence factors, and phylogroups of *E. coli* isolates during the killing process in a bovine slaughterhouse. We analyzed 336 samples (from water, surfaces, carcasses, and feces), and 83.3% (280/336) were positive for *E. coli*. The most common phenotypic resistances that we detected were 50.7% (142/280) for tetracycline, 44.2% (124/280) for cephalothin, 34.6% (97/280) for streptomycin, and 36.7% (103/280) for ampicillin. A total of 82.4% of the isolates had resistance for at least one antimicrobial, and 37.5% presented multiresistance. We detected a total of 69 different phenotypic resistance patterns. We detected six other resistance-related genes, the most prevalent being *tet*A (22.5%) and *str*B (15.7%). The prevalence values of the virulence genes were 5.4% in *hly*A, 1.4% in *stx*1, and 0.7% in *stx*2. The frequencies of the pathogenic strains (B2 and D) were 32.8% (92/280) and 67.1% (188/280) as commensals A and B1, respectively. *E. coli* isolates with pathogenic potential and multiresistance may represent an important source of dissemination and a risk to consumers.

## 1. Introduction

The indiscriminate use of antibiotics in cattle to promote growth and prevent diseases is directly related to the incidence of resistant bacteria in their resident microbiota, which could eventually contaminate the carcass during the slaughter process or subsequent manipulation and be transmitted to consumers and distributed in the environment [1,2,3,4]. Although the Mexican Animal Health Federal Law of 2007 approved the minimal use of antimicrobials in cattle and only with a veterinary prescription [5], little is known about the antimicrobial-resistant strains present in animals for human consumption in Mexico. Maradiaga et al. [5] analyzed the antimicrobial susceptibility in *Salmonella* isolates from cattle and beef from southeastern Mexico, and according to the results, 58.4% of the strains were resistant to at least one antimicrobial drug. In the northeast of Mexico, susceptibility pattern information is scarce or not available. Martinez-Vázquez et al. (2021) [6] studied the antimicrobial resistance profiles of *Escherichia coli* strains isolated from bovine feces and carcass samples from Tamaulipas, Mexico. According to the results, 94.8% of the strains had resistance to at least one antimicrobial, and 72.7% were multidrug-resistant.

Additionally, antimicrobial resistance is commonly associated with virulence factors. Researchers described two genes on large transferable hybrid plasmids in pathogenic *E. coli* isolates recovered from animals [7,8,9]. Combinations of virulence factors allow for the division of the *E. coli* strain clusters into phylogenetic groups, and this can be used to determine potential health risks [10]. The objective of this study was to determine the susceptibility patterns, coexistence of resistance genes, virulence factors, and phylogenetic clustering of *E. coli* isolates obtained in the process of sacrificing cattle in a slaughterhouse from the central zone of Tamaulipas, Mexico.

## 2. Results

### 2.1. Prevalence of Escherichia coli

A total of 83.3% (280/336) of the samples were positive for *E. coli* (Table 1). The prevalence of *E. coli* was 50% (30/60) in the water samples (8.3% in carcass washing water; 33.3% in cistern water; 100% in discharge water); 41.6% (15/36) on surfaces (25% in the process area; 33.3% in carcass transportation trucks; 66.6% on cold room frames and doors); 97.5% (117/120) in carcasses; 98.3% (118/120) in feces.

### 2.2. Antimicrobial Susceptibility Profiles

The most common phenotypic resistance that we detected was for tetracycline with 50.7% (142/280), followed by cephalothin with 44.2% (124/280), streptomycin with 34.6% (97/280), ampicillin with 36.7% (103/280), trimethoprim–sulfamethoxazole with 16% (45/280), and chloramphenicol with 14.2% (40/280). A total of 100% (280/280) of the strains were sensitive to netilmicin, 99.6% (279/280) were sensitive to ceftriaxone and levofloxacin, 97.1% (272/280) were sensitive to gentamicin, and 92.5% (259/280) were sensitive to cefepime. For cefotaxime and ceftriaxone, we did not detect any resistant strains at the intermediate level. Only 3.5% (10/280) of the isolates had sensitivity to all the drugs. By sample types, the carcass and fecal isolates had higher resistance levels, followed by the water and surface samples. Overall, 82.8% (232/280) of the total isolates had resistance for at least one antimicrobial, and of these, 37.5% (105/280) had resistance for three or more (multiresistant) antimicrobials, according to the formula proposed by Selim et al. [11]. Thus, of the total multiresistant isolates, we detected 4.2% (48/280) in the carcass samples, 38.4% (45/117) in the feces, 7.6% (9/118) in the water, and 10.0% (3/30) on surfaces (Table 2). We detected a total of 69 different phenotypic resistance patterns (Table 3). Finally, 3.9% (11/280) of the isolates with phenotypic resistance presented virulence factors, and of these, 1.4% (4/280) were multiresistant. Moreover, 67.1% (188/280) of the isolates with resistance profiles corresponded to the phylogroups categorized as commensals (A and B1), and 32.8% (92/280) corresponded to pathogens (B2 and D).

### 2.3. Prevalence of Resistance-Related Genes

We detected six different resistance-related genes in the total population of *E. coli* isolates (*n* = 280) (*tet*A; *tet*B; *aac*(3); *aad*A; *str*A; *str*B). We detected resistance genes in 51% (143/280) of the isolates. Of these, we detected one gene in 74.1% (/106/143), two genes in 15.3% (22/143), three genes in 8.3% (12/143), and four genes in only 1.3% (2/143). None of the isolates harbored the six genes studied. We detected the *tet*A gene the most, with a prevalence of 22.5% (63/280), and mostly in the carcass samples. We detected the *str*B and *tet*B genes in the four types of samples at 15.7% (44/280) and 8.9% (25/280), respectively. The frequencies of the *str*A (2.1%; 6/280) and *aad*A (1.7%; 5/280) genes were low in the carcasses, feces, and water isolates, and they were absent in the surface samples. We did not detect the *aac*(3) gene in any of the isolates (Table 4).

### 2.4. Prevalence of Virulence-Related Genes

We detected at least one virulence gene in 6.7% (19/280) of the *E. coli* isolates. We detected the *hly*A gene in 5.3% (15/280) of the isolates, eight of which we detected in the water samples (26.6%; 8/30) (seven in the abdominal viscera and disgorge discharge water samples and one from cistern water), five in the stool samples (4.2%; 5/118), one in a carcass sample (0.8%; 1/117), and one in a surface sample (6.6%; 1/15) (from the cold room door and frame area). We detected the *stx*1 gene in three isolates (1.0%; 3/280): two from carcass samples and one from a water sample (the abdominal viscera discharge area). We only detected the *stx*2 gene in one carcass sample (0.3%; 1/280). We detected the *stx*1 and *stx*2 genes in three strains isolated from carcass samples and one from water; however, we did not detect any of these genes in the isolates from the surface or fecal samples. We did not detect more than two virulence genes in any of the analyzed strains (Table 1). 

### 2.5. Phylogenetic Group Frequencies

We classified the *E. coli* isolates (280) into the four phylogenetic groups proposed by Clermont et al. [12]. Overall, we classified 49.6% (139/280) of the strains in Phylogroup A, 17.5% (49/280) in Phylogroup B1, 15.7% (44/280) in Phylogroup B2, and 17.1% (48/280) in Phylogroup D (Table 1). A total of 32.8% (92/280) of the strains from the phylogroups were pathogenic (B2 and D), and 67.1% (188/280) of the isolates were commensals (A and B1). The fecal (15.7% (44/280)) and carcass (12.5% (35/280)) isolates made up the highest percentage of pathogenic phylogroups (B2 + D), followed by the water samples (4.2% (12/280)) and surfaces (0.3% (1/280)).

Finally, the results obtained for the virulence, resistance to antibiotics, and distribution in the phylogroups were associated (Table 5).

## 3. Discussion

The high prevalence of *E. coli* detected in the carcass samples (97.2%) reflects the inadequate practices in the slaughter process, either due to fecal cross-contamination or improper meat handling, which may represent a latent public health risk. The prevalence values detected were slightly higher than the results published by Martínez-Vázquez [6] (83.8% in carcasses from Tamaulipas); however, they are similar to those reported by Martínez-Chavez et al. [13] (97.1%) in a study conducted on three municipal slaughterhouses in Jalisco, Mexico. In Arusha (Tanzania), Murutu et al. [14] detected *E. coli* in 70% of the studied carcasses, which is a lower percentage than the one detected in this study (97.2%), with a similar number of analyzed carcasses. The detected prevalence in this study was high compared with those of other similar studies (South Korea: 47% [15]; Australia: 25% [16]; northwestern Mexico: 13.4% [17]). The presence of *E. coli* in 33.3% of the cistern water samples, although treated by chlorination before use in the slaughterhouse, demonstrated that this treatment requires more attention to minimize the microbiological risks that may reach the carcass and process surfaces. However, the inappropriate and irrational use of antimicrobials is one of the leading causes of the increase in bacterial drug resistance, as well as one of the most critical current public health problems [18]. In veterinary medicine, the use of antimicrobial agents is a common practice in the therapy of various pathologies, either prophylactically or as growth promoters in production animals, which is a situation that favors the presence of resistant bacterial strains [19,20]. In this study, the highest phenotypic resistances that we detected were 50.7% for tetracycline, followed by 44.2% for cephalothin, and 36.7% for ampicillin. For a similar study conducted in Tamaulipas, Mexico, Martínez-Vázquez et al. [6] report values higher than those that we detected, with 69.0% for tetracycline, 76.0% for cephalothin, and 83.0% for ampicillin in fecal and carcass bovine isolates. Although both studies were conducted in the same area of Tamaulipas, the results may vary because the samplings were two years apart and from different farms. The criteria for managing and consuming antibiotics are different on each farm, depending on the owner. Since 2010, measures have been applied in Mexico to regulate the sale and dispensation of antibiotics, which are only by prescription and when issued by health professionals [21]. However, the high percentages of antibiotic-resistant strains observed in this study suggests that the indications might not be respected and that there is an abuse of the use of antibiotics. In a previous study from the same state, Martínez-Vázquez et al. [22] analyzed commercial ground beef samples, reporting resistance percentages higher than those of the current work, with resistances of 67.7% for tetracycline, 91.8% for cephalothin, and 90.5% for ampicillin. We acquired the ground beef samples from stores and they did not have identified origins; thus, they could have been mixtures of meat from several farms in different areas. In a study conducted by Aguilar-Montes et al. [23], the authors detected 36% resistance to ampicillin in bovine carcass isolates in the State of Mexico and Jalisco (Mexico), which is a similar result to that in this study. The carcass and fecal samples from the studied cattle were from different farms in the region, and mainly from extensive systems (grazing); thus, there was no information on the animals before slaughter, and it was not possible to associate the presence of isolates with high-resistance profiles with the sanitary management of specific livestock farms. The *E. coli* strains recovered from the four types of samples (water, carcasses, surfaces, and feces) exhibited high phenotypic resistance (82.4%) and multiresistance (37.5%). According to these results, bovines may represent an important reservoir of antibiotic-resistant bacteria, are a potential route of their transmission, and represent a health risk. 

Overall, we detected antimicrobial resistance genes in 37% (104/280) of the isolates (*tet*A, *tet*B, *str*A, *str*B, *aad*A, and *aac*(3)). Of the 142 *E. coli* strains with phenotypic resistance to tetracycline, we detected 5.6% (8/142) with both tested tetracycline resistance genes (*tet*A and *tet*B), and 56.3% (80/142) with one of the two resistance genes. Of the 97 strains with phenotypic resistance to streptomycin, 56.7% (55/97) had one or various resistance genes (*str*A (6.1% (6/97)), *str*B (45.3 (44/97)), or *aad*A (5.1% (5/97))). We did not detect the *aac(3)* gene in any of the isolates. We observed a positive association between the presence of resistance genes and the phenotypic resistances for tetracycline (*p* ˂ 0.001) and streptomycin (*p* ˂ 0.01) (Table 5). In a study conducted by Ayaz et al. [24] on *E. coli* isolates from slaughterhouse cattle and wastewater, the authors did not find any correlation between the phenotypic resistance and presence of resistance genes. According to the high-resistance and multiresistance profiles detected in most of the isolates, we need to consider testing more genes that are associated with drug resistance on other antimicrobials (such as cephalosporins and beta-lactams), which would allow for a more precise estimation of the association or independence between the isolates that present resistance factors and those that present phenotypic resistance profiles.

In this study, we detected three virulence factors with prevalence values of 6.7% (19/280). A total of 47.3% (9/280) of the 19 isolates presented virulence factors and phenotypic resistance profiles, and 15.7% (3/280) displayed multiresistance. The two isolates that presented the two virulence genes *stx*1 and *stx*2 did not present any phenotypic resistance. The three multiresistant isolates harbored the *hly*A gene. Additionally, 2.5% (7/280) of the isolates that presented virulence factors harbored at least one of the studied resistance genes. Only one isolate presented four of the six studied genes (*tet*A, *tet*B, *str*A, and *str*B). Four strains presented virulence genes, resistance genes, and phenotypic resistance (two presented resistance and two presented multiresistance). The prevalence values of the studied virulence factor genes were low compared with those of similar studies. The *hly*A gene was the most frequently detected in this study, with a prevalence of 5.4%, which is a higher value than that published by Martínez-Vázquez et al. [6] (2.8%). In the current study, we observed the prevalence of the *stx*1 and *stx*2 genes at only 1.4% and 0.7%, respectively, while Martínez-Vázquez et al. [6] report higher values: 4.2% for the *stx*1 gene and 7.0% for the *stx*2 gene (1.9%). Moreover, the values are different (higher prevalence) than the ones reported by Li et al. [25] (*stx*1: 4.9%; *stx*2: 27.6%) in a study conducted on various retail food products sold in China. Minh et al. [26] also report higher prevalence values (*stx*1: 6.6%; *stx*2: 14.8%), as do Ateba and Mbwe [27] (*stx*1: 6.2%; *stx*2: 17.5%). The total and fecal coliform counts in the water samples, coupled with the detection of *E. coli* and virulence gene isolates (*hly*A), suggest that a possible source of contamination of the carcasses and surfaces is the water used during the slaughterhouse cleaning processes. In addition, although the isolates detected with the *stx*1 and *stx*2 genes were low, and we only detected both genes in two isolates of the carcass samples, there is an evident risk of transmitting potentially pathogenic strains [28]. We classified the total isolates (280) into four phylogenetic groups: 32.8% (92/280) were pathogenic (B2 and D), and 67.1% (188/280) were commensals (A and B1). We detected the highest percentage of pathogenic groups in the stool samples (38%), followed by the water (37%), carcass (31%), and surface (7%) samples. Among the isolates that we phylogrouped as pathogens, 8.6% (8/93) presented virulence factors, 41.9% (39/93) presented resistance genes, 80.6% (75/93) presented resistance profiles, and 35.5% (33/93) presented multiresistance. Morcatti et al. [29] report different results from those of this study for isolates from dairy cattle feces in Brazil: they detected 4.6% in Phylogroup A, 74% in Phylogroup B1, 0.6% in Phylogroup B2, and none in Phylogroup D. However, in dairy cattle isolates from Zambia, Mainda et al. [30] detected 9% for Phylogroup A, 67% for Phylogroup B1, 4% for Phylogroup B2, and 9% for Phylogroup D. According to these two studies, the B1 group is the majority, which is contrary to our study (22%), in which Phylogroup A was higher (47%). Taking the isolates as the reference from which we detected the virulence factors (6.8%; 19/280), we detected resistance genes in 36.8% (7/19), phenotypic resistance in 57.8% (11/19), and multiresistance in 21.1% (4/11). Of these, 42.1% (8/19) were in the pathogenic phylogroups (B2 and D). When relating these three conditions, 15% (3/19) presented virulence genes and resistance genes and were phylogrouped as pathogens, with one per type of sample, except for the surface samples. From the water and feces samples, two isolates presented the four studied factors: one with resistance and one with multiresistance.

According to the X^2^ analysis of the association among the isolates that presented virulence factors (independent variable) according to the phylogroups (dependent variable), the isolates grouped as virulent (Phylogroups B2 and D) statistically do not depend on the presence of the studied virulence genes (*stx*1, *stx*2, and *hly*A) (*p* > 0.05) (Table 5). It is essential to use a more substantial number of virulence genes of the studied strains to broaden our understanding of the association levels among the pathogenic phylogroups concerning the presence of virulence factors.

## 4. Materials and Methods

### 4.1. Sample Collection 

We performed the sampling in a federal-inspection-type slaughterhouse (TIF) for bovines in northeastern Mexico over one year.

Water. We analyzed the following: (a) treated (chlorinated) water for carcass washing; (b) untreated water (cistern); (c) discharge water from the abdominal viscera wash area; (d) discharge water from the disgorge area. We obtained 500 mL of water sample in a sterile glass bottle from each sampling point. The sampling frequency for the treated water was 15 days (*n* = 24), and it was monthly for the other three samples (12 of each sample type; *n* = 36).

Carcasses. We conducted the sampling following NOM-109-SSA1-1994. After the slaughter, we maintained the carcasses for at least twelve hours in the refrigeration chamber, and we selected them by simple random sampling. We performed the sampling using sterile sponges (Speci-Sponge^®^, Nasco Whirl-Pak) that we moistened with buffered peptone water (BPW). From each carcass, we sampled three anatomical site surfaces: the skirt, brisket, and perianal regions. We covered a surface of 100 cm^2^ at each point (300 cm^2^ per carcass), and we then placed the sponge in a sterile plastic bag and added 25 mL of BPW. We took five carcass samples per sampling, at a biweekly frequency (*n* = 120).

Surfaces. We performed the same technical process for the surface sampling as we did for the carcasses. We analyzed three surface areas: (1) cold room doors and frames; (2) process area walls; (3) carcass transportation trucks. The sampled surface from each area was five nearby points, and each covered a surface area of 100 cm^2^ (500 cm^2^ of total surface area). The sampling frequency was monthly (12 of each area (*n* = 36)).

Stool. We performed the stool sampling in the abdominal viscera wash area. We took approximately 100 g of feces from the distal area of the large intestine using a sterile polyethylene bag. We took five samples every 15 days (*n* = 120).

### 4.2. Isolation and Identification of Escherichia coli 

We inoculated all the samples (water, carcasses, surfaces, and feces) in lactose broth (LB) (BD Bioxon, Cat. 211700) at a 1:9 ratio, and we incubated them at 37 °C for 24 h. We then seeded the lactose broth on eosin methylene blue (EMB) agar plates (DIBICO, Cat. 1011), which we incubated at 37 °C from 18 to 24 h. From each plate, we selected three colonies with the presumptive morphological characteristics of *E. coli*. We inoculated the isolates on tryptic soy agar (TSA) plates (BD Bioxon, Cat. 210800), and we grew them at 37 °C for 24 h to confirm their purity. We performed the isolate identification with biochemical assays for lactose and glucose fermentation in triple sugar iron (TSI) agar (BD Bioxon, Cat. 211400), sulfide–indole–motility (SIM) medium (BD Bioxon, Cat. 210100), Methyl Red–Voges–Proskauer (MR-VP) with broth (BD Bioxon, Cat. 211691), and Simmons’ citrate (SC) (BD Bioxon, Cat. 211761). We incubated all the isolates at 37 °C for 24 h, following the indications described in the Cowan and Steel’s Manual [31]. For the PCR identification, we obtained the bacterial DNA by suspending the bacterial colonies from a fresh culture boiled at 95 °C for 15 min [6]. We performed the PCR analyses using specific primers (*mdh*): F 5′GGTATGGATCGTTCCGACCT 3′ and R 5′GGCAGAATGGTAACACCAGAGT 3′ [32]. The PCR reaction mixture contained 1X buffer, 25 mM MgCl_2_, 10 mM dNTPs, 10 mM primers, 5 U Taq DNA polymerase, and sterile water, for a final volume of 25 µL. The PCR amplification conditions were initial denaturation at 95 °C for 1 min, followed by 30 cycles of denaturation at 95 °C for 45 s, annealing at 53 °C for 45 s, extension at 72 °C for 45 s, and a final amplification cycle at 72 °C for 7 min. We observed the PCR products on 2.0% agarose gels at 100 V for 45 min.

### 4.3. Antimicrobial Susceptibility Testing

We performed the susceptibility tests according to the Kirby–Bauer method and following the Clinical and Laboratory Standards Institute [33]. We tested 16 antimicrobials in individual antimicrobial susceptibility discs (BD BBL): amikacin (AN; 30 µg); ampicillin (AM; 10 µg); amoxicillin–clavulanic acid (AMC; 30 and 10 µg); cephalothin (CF; 30 µg); cefepime (FEP; 30 µg); cefotaxime (CTX; 30 µg); ceftriaxone (CRO; 30 µg); ciprofloxacin (CIP; 5 µg); chloramphenicol (C; 30 µg); streptomycin (S; 10 µg); gentamicin (GM; 10 µg); netilmicin (NET; 30 µg); nitrofurantoin (FM; 100 µg); levofloxacin (LVX; 5 µg); trimethoprim/sulfamethoxazole (SXT; 1.25 and 23.75 µg); tetracycline (TE; 30 µg). We classified the inhibition zones into three categories according to the NCCLS: Resistant (R), Intermediate (I), and Sensitive (S).

### 4.4. Detection of Antimicrobial Resistance Genes

For the DNA extraction from the *E. coli* isolates, we exposed the bacterial biomasses from the TSA plates to a thermal shock of 95 °C in 500 µL of sterile MiliQ water for 15 min to obtain the lysates. We analyzed the presence of the genes associated with the resistance to tetracycline (*tet*(A) and *tet*(B)) and aminoglycosides (*aac*(3)IV, *aad*A, *str*A, and *str*B) using PCR (Ng et al. and Kosak et al. [34,35]). The PCR reaction mixture was composed of 1X buffer, 25 mM MgCl_2_, 10 mM dNTPs, 10 mM primers, 5 units of Taq DNA polymerase, and sterile water, for a final volume of 25 µ. We used the positive and negative controls from the collection of the Instituto Politécnico Nacional obtained in previous studies for each test. We visualized the PCR products via electrophoresis on 2.7% agarose gels with SYBR Green and a 100 bp-molecular-weight marker at 100 volts for 40 min.

### 4.5. Virulence-Related Gene Detection

We performed virulence-related gene identification following the protocols proposed by Canizalez et al. [17]. We identified three genes (*stx*1, *stx*2, and *hly*A). The volume and concentration of the PCR reaction mixture were the same as those used for the resistance genes. We visualized the PCR products via electrophoresis on 2% agarose gels at 100 volts for 40 min. We performed the electrophoresis by comigrating a 100 bp-molecular-weight marker, and we used SYBR Green to stain the amplified products.

### 4.6. Phylogenetic Group Classification

We classified the *E. coli* strains isolated in this study using the method proposed by Clermont et al. [12], which allows for typing the strains that do not correspond with their virulence values. We determined the presence of the *chu*A (288 bp), *yja*A (211 bp), and *Tsp*E4C (152 bp) genes with a triplex PCR performed on the lysates. We performed the amplicon visualization using gel electrophoresis on 2% agarose gels. To interpret the presence or absence of the bands that correspond to the studied genes and fragments, we used the methodology of Clermont et al. [12], which classifies them into four phylogenetic groups: A, B1, B2, and D.

### 4.7. Statistical Analysis

We performed a parametric statistical analysis (relative frequencies) and statistical analysis of the association by X^2^ to the variables of phenotypic resistance, resistance genes, virulence genes, and phylogroups.

## 5. Conclusions

The high prevalence of *E. coli* detected in the slaughter process could reflect inefficient management practices, and it highlights the need for monitoring to identify the critical points of contamination that need improvement. Moreover, the high percentages of strains with antimicrobial resistance (82.4%) and multiresistance (37.5%) indicate that the cattle could have been exposed to the inappropriate or excessive use of antibiotics during breeding. Although the virulence factors of the analyzed strains had low prevalence in this study (32.8%), because they are classified in the B2 and D phylogroups, they could become a risk to the health of consumers. According to the results, the bovines included in this study are an important reservoir of antibiotic- and pathogen-resistant bacteria, acting as a means of propagation.

## Figures and Tables

**Table 1 antibiotics-12-00291-t001:** Phylogenetic group distributions and virulence factor prevalence values of *E. coli* isolated from water, surface, channel, and fecal samples.

Sample Type	Sample Number	*E. coli* Prevalence (*n* (%))	Phylogenetic Groups (*n* (%))				Virulence Factors (*n* (%))		
			A	B1	B2	D	*stx*1	*stx*2	*hlyA*
Water	60	30 (50)	6 (20)	13 (43)	9 (30)	2 (7)	1 (3.3)	0 (0)	8 (26.6)
Surfaces	36	15 (41.7)	14 (93)	0 (0)	1 (7)	0 (0)	0 (0)	0 (0)	1 (6.6)
Carcasses	120	117 (97.5)	55 (47)	26 (22)	17 (15)	19 (16)	2 (1.7)	1 (0.8)	1 (0.8)
Feces	120	118 (98.3)	63 (53)	10 (8)	19 (16)	26 (22)	0 (0)	0 (0)	5 (4.2)
Total	336	280 (83.3)	138 (49)	49 (18)	46 (16)	47 (17)	3 (1.0)	1 (0.3)	15 (5.3)

**Table 2 antibiotics-12-00291-t002:** Resistance and multiresistance profiles of *E. coli* to 16 antimicrobials, isolated from water, surfaces, carcasses, and feces (*n* = 280).

Antimicrobial	Water (*n* = 30)		Surfaces (*n* = 15)		Carcasses (*n* = 117)		Feces (*n* = 118)	
	R	S	R	S	R	S	R	S
	Isolates (*n* (%))							
Amikacin	1 (3.3)	28 (93.3)	0 (0)	15 (100)	0 (0)	111 (95.7)	7 (5.9)	95 (80.5)
Ampicillin	9 (30)	18 (60)	3 (20)	11 (93.3	38 (32.8)	45 (38.8)	51 (43.2)	44 (37.3)
Amoxicillin/clavulanic acid	0 (0)	24 (80)	0 (0)	14 (93.3)	12 (10.3)	65 (56)	6 (5.1)	74 (62.7)
Cephalothin	6 (20)	10 (33.3)	3 (20)	4 (26.7)	57 (49.1)	13 (11.2)	64 (54.2)	13 (11)
Cefepime	0 (0)	29 (96.7)	0 (0)	15 (100)	0 (0)	109 (94)	1 (0.8)	102 (86.4)
Cefotaxime	0 (0)	26 (86.7)	0 (0)	15 (100)	0 (0)	94 (81)	0 (0.0)	93 (78.8)
Ceftriaxone	0 (0)	30 (100)	0 (0)	15 (100)	0 (0)	115 (99.1)	0 (0.0)	117 (99.2)
Ciprofloxacin	0 (0)	30 (100)	1 (6.7)	14 (93.3)	2 (1.7)	114 (98.3)	0 (0.0)	117 (99.2)
Chloramphenicol	4 (13.3)	25 (83.3)	2 (13.3)	13 (86.7)	18 (15.5)	97 (83.6)	13 (11.0)	102 (86.4)
Streptomycin	9 (30)	20 (66.7)	4 (26.7)	10 (66.7)	61 (52.6)	41 (35.3)	37 (31.4)	55 (46.6)
Gentamicin	0 (0)	28 (93.3)	0 (0)	15 (100)	1 (0.9)	114 (98.3)	3 (2.5)	113 (95.8)
Netilmicin	0 (0)	30 (100)	0 (0)	15 (100)	0 (0)	116 (100)	1 (0.8)	116 (98.3)
Nitrofurantoin	5 (16.7)	18 (60)	1 (6.7)	14 (93.3)	3 (38.8)	68 (58.6)	6 (5.1)	77 (65.3)
Levofloxacin	0 (0)	30 (100)	1 (6.7)	14 (93.3)	1 (0.9)	115 (99.1)	0 (0.0)	118 (100.0)
Trimethoprim/sulfamethoxazole	6 (20)	24 (80)	3 (20)	12 (80)	18 (15.5)	93 (80.2)	18 (15.3)	97 (82.2)
Tetracycline	14 (46.7)	14 (46.7)	6 (40)	8 (53.3)	61 (52.6)	48 (41.4)	62 (52.5)	53 (44.9)

Note: Percentage differences correspond to intermediate profiles. R: resistant; S: susceptible.

**Table 3 antibiotics-12-00291-t003:** Some resistance profiles in *E. coli* isolated from water, surface, carcass, and fecal samples.

Patterns of Resistance Phenotypes											
P50		CF	AM					NF			
P51	TE	CF	AM	STR	STX			NF			
P52	TE		AM	STR	STX			NF			
P53	TE		AM		STX			NF			
P54	TE		AM	STR		CL		NF			
P59	TE	CF			STX				AMC		
P60	TE	CF	AM	STR					AMC		
P61	TE		AM		STX				AMC		
P62	TE	CF	AM			CL			AMC		
P63	TE		AM		STX	CL			AMC		
P64	TE	CF	AM	STR		CL			AMC		
P65	TE	CF	AM	STR					AMC		
P66	TE	CF	AM	STR	STX		GE		AMC		
P67	TE	CF	AM	STR	STX				AMC		
P68		CF								CIP	
P69	TE	CF	AM	STR	STX					CIP	LEV

Note: TE: tetracycline; CF: cephalothin; AM: ampicillin; STR: streptomycin; STX: trimethoprim/sulfamethoxazole; CL: chloramphenicol; GE: gentamicin; NF: nitrofurantoin; AMC: amoxicillin–clavulanic acid; CIP: ciprofloxacin; LEV: levofloxacin.

**Table 4 antibiotics-12-00291-t004:** Resistance-related genes in *E. coli* isolated from water, surface, carcass, and fecal samples.

Sample Type	Number of Isolates	Resistance-Related Genes (*n* (%))					
		*tet*A	*tet*B	*aac*(3)	*aad*A	*str*A	*str*B
Water	30	7 (23.3)	2 (6.6)	0 (0)	1 (3.3)	3 (10)	6 (20)
Surfaces	15	2 (13.3)	1 (6.6)	0 (0)	0 (0)	0 (0)	1 (6.6)
Carcasses	117	34 (28.8)	15 (12.8)	0 (0)	3 (2.5)	2 (1.7)	26 (22.2)
Feces	118	20 (16.9)	7 (5.9)	0 (0)	1 (0.8)	1 (0.8)	11 (9.3)
Total	280	63 (22.5)	25 (8.9)	0 (0)	5 (1.7)	6 (2.1)	44 (15.7)

**Table 5 antibiotics-12-00291-t005:** Frequencies, percentages, and *p*-values for association of virulence genes, resistance, and phylogroups in *E. coli*.

	Pathogens		Commensals		Total		
	Frequency	Percentage	Frequency	Percentage	Frequency	Percentage	*p*-Value
Virulence genes							*p* > 0.05
Positive isolates	7	38.9	11	61.1	18	100	
Negative isolates	86	32.8	176	67.2	262	100	
Tetracycline-resistant genes							*p* > 0.05
Positive isolates	30	37.0	51	63.0	81	100	
Negative isolates	63	31.7	136	68.3	199	100	
Streptomycin-resistant genes							*p* < 0.05
Positive isolates	21	46.7	24	53.3	45	100	
Negative isolates	72	30.6	163	69.4	235	100	

## Data Availability

The data are available upon request.

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
