# Peer review of "Molecular and Antimicrobial Susceptibility Characterization of Escherichia coli Isolates from Bovine Slaughterhouse Process"

_antibiotics, 2023, doi:10.3390/antibiotics12020291_

Round 1

Reviewer 1 Report

Given the importance of antibiotic resistance, I would like to congratulate my colleagues on this work. The paper is generally well-written and structured. However, I would recommend that the authors carefully revise the suggestions below: 

1. First of all, information about multidrug resistance is missing (molecules): I would suggest including a table that better highlights the phenomenon of MDR, not only the percentage but also antibiotics should be highlighted. In my opinion, in this context, should be included also the Antimicrobial Resistance Pattern.

2. In the discussion, I would like to see a paragraph or comment on the use of antibiotics in Mexico.

3. Line 17:  You said "The present work aims to determine the susceptibility patterns, virulence factors, 16 and phylogroups of E. coli isolates in the killing process in a bovine slaughterhouse" Where? I would suggest including in this sentence, also the locality or country where this research is being conducted.

Thank you.

Reviewer 2 Report

this study entitled "Molecular and antimicrobial susceptibility characterization of Escherichia coli isolates from a bovine slaughterhouse process" is well-planned and analyzes the antibiotic resistance, phylotypes, virulence, and antibiotic resistance genes in E. coli isolated from the bovine slaughterhouse environment. The study has limitations and should be improved as per the following comments and the attached reviewed file before being considered for publication.

1) Study describe the isolation from one specific slaughterhouse, how the results can be generalized, or how the results would be of interest to readers. The addition of data on slaughterhouse capacity and importance in the country can add merit to the study. 

2) E. coli are identified by only biochemical techniques. i think molecular identification by PCR is also important to be added.

3) On what basis the list of tested antibiotics is prepared? some of the antibiotics are not used for veterinary applications.

4)   Why only a few specific virulence genes and antibiotic resistance genes are determined? What was the rationale behind that choice?

5) Introduction and discussion need to structured better

6) MLST or modified Clermont method is more suitable for phylogenetic typing as it discriminates E. coli into 8 phylotypes.

Reviewer 3 Report

The article entitled "Molecular and antimicrobial susceptibility characterization of Escherichia coli isolates from a bovine slaughterhouse process" highlight the prevalence and identification of AMR phenotypically and genotypically in E. coli isolated from the beef slaughterhouse in Mexico.

Although the study is good and has scientific significance for the scientific community and readers and alerting the consumers and scientific community regarding public health issues associated with AMR E. coli strains. However, the writeup and methodology need significant improvement. I have a few comments as given below;

1. The manuscript writing is not suitable for this high-rated journal. I suggest authors consult some good English speakers or at least use English editing services for an efficient writeup.

2. Other than language editing and improvement, the manuscript also needs improvement in scientific writing. 

3. In the methodology section, subsection 4.4. detection of AMR genes, the authors mentioned that they used positive and negative controls. I suggest adding the ID No. of these controls along with references. 

4. In Section 4.6 in Methodology, the authors performed the Phylogenetic classification of groups, but they didn't mention or explained that they are performing this classification for which organism and for what purpose.

5. Conclusion is not fully supporting the results or findings of this study. I suggest adding more to support your study.  

Thanks 

Round 2

Reviewer 3 Report

Dear Authors, 

Thank you, I think now the article is in better shape. 

Author Response

Dear Reviewer,

Thanks for helping us to improve the paper.

Best regards.